# GLOBAL SELF-ATTENTION NETWORKS FOR IMAGE RECOGNITION

## ABSTRACT

Recently, a series of works in computer vision have shown promising results on various image and video understanding tasks using self-attention. However, due to the quadratic computational and memory complexities of self-attention, these works either apply attention only to low-resolution feature maps in later stages of a deep network or restrict the receptive field of attention in each layer to a small local region. To overcome these limitations, this work introduces a new global self-attention module, referred to as the GSA module, which is efficient enough to serve as the backbone component of a deep network. This module consists of two parallel layers: a content attention layer that attends to pixels based only on their content and a positional attention layer that attends to pixels based on their spatial locations. The output of this module is the sum of the outputs of the two layers. Based on the proposed GSA module, we introduce new standalone global attention-based deep networks that use GSA modules instead of convolutions to model pixel interactions. Due to the global extent of the proposed GSA module, a GSA network has the ability to model long-range pixel interactions throughout the network. Our experimental results show that GSA networks outperform the corresponding convolution-based networks significantly on the CIFAR-100 and ImageNet datasets while using less parameters and computations. The proposed GSA networks also outperform various existing attention-based networks on the ImageNet dataset.

## 1 INTRODUCTION

Self-attention is a mechanism in neural networks that focuses on modeling long-range dependencies. Its advantage in terms of establishing global dependencies over other mechanisms, e.g., convolution and recurrence, has made it prevalent in modern deep learning. In computer vision, several recent works have augmented Convolutional Neural Networks (CNNs) with global self-attention modules and showed promising results for various image and video understanding tasks (Bello et al., 2019; Chen et al., 2018; Huang et al., 2019; Shen et al., 2018; Wang et al., 2018; Yue et al., 2018). For brevity, in the rest of the paper, we refer to self-attention simply as attention.

The main challenge in using the global attention mechanism for computer vision tasks is the large spatial dimensions of the input. An input image in a computer vision task typically contains tens of thousands of pixels, and the quadratic computational and memory complexities of the attention mechanism make global attention prohibitively expensive for such large inputs. Because of this, earlier works such as Bello et al. (2019); Wang et al. (2018) restricted the use of global attention mechanism to low-resolution feature maps in later stages of a deep network. Alternatively, other recent works such as Hu et al. (2019); Ramachandran et al. (2019); Zhao et al. (2020) restricted the receptive field of the attention operation to small local regions. While both these strategies are effective at capping the resource consumption of attention modules, they deprive the network of the ability to model long-range pixel interactions in its early and middle stages, preventing the attention mechanism from reaching its full potential.

Different from the above works, Chen et al. (2018); Huang et al. (2019); Shen et al. (2018); Yue et al. (2018) made the global attention mechanism efficient by either removing the softmax normalization on the product of queries and keys and changing the order of matrix multiplications involved in the attention computation (Chen et al., 2018; Shen et al., 2018; Yue et al., 2018) or decomposing

one global attention layer into a sequence of multiple axial attention layers (Huang et al., 2019). However, all these works use content-only attention which does not take the spatial arrangement of pixels into account. Since images are spatially-structured inputs, an attention mechanism that ignores spatial information is not best-suited for image understanding tasks on its own. Hence, these works incorporate attention modules as auxiliary modules into standard CNNs.

To address the above issues, we introduce a new global self-attention module, referred to as the GSA module, that performs attention taking both the content and spatial positions of the pixels into account. This module consists of two parallel layers: a content attention layer and a positional attention layer, whose outputs are summed at the end. The content attention layer attends to all the pixels at once based only on their content. It uses an efficient global attention mechanism similar to Chen et al. (2018); Shen et al. (2018) whose computational and memory complexities are linear in the number of pixels. The positional attention layer computes the attention map for each pixel based on its own content and its relative spatial positions with respect to other pixels. Following the axial formulation (Ho et al., 2019; Huang et al., 2019), the positional attention layer is implemented as a column-only attention layer followed by a row-only attention layer. The computational and memory complexities of this axial positional attention layer are $O(N\sqrt{N})$ in the number of pixels.

The proposed GSA module is efficient enough to act as the backbone component of a deep network. Based on this module, we introduce new standalone global attention-based deep networks, referred to as global self-attention networks. A GSA network uses GSA modules instead of convolutions to model pixel interactions. By virtue of the global extent of the GSA module, a GSA network has the ability to model long-range pixel interactions throughout the network. Recently, Wang et al. (2020) also introduced standalone global attention-based deep networks that use axial attention mechanism for both content and positional attentions. Different from Wang et al. (2020), the proposed GSA module uses a non-axial global content attention mechanism that attends to the entire image at once rather than just a row or column. Our experimental results show that GSA-ResNet, a GSA network that adopts ResNet (He et al., 2016) structure, outperforms the original convolution-based ResNet and various recent global or local attention-based ResNets on the widely-used ImageNet dataset.

MAJOR CONTRIBUTIONS

- **GSA module:** We introduce a new global attention module that is efficient enough to act as the backbone component of a deep network. Different from Wang et al. (2018); Yue et al. (2018); Chen et al. (2018); Shen et al. (2018); Huang et al. (2019), the proposed module attends to pixels based on both content and spatial positions. Different from Zhao et al. (2020); Hu et al. (2019); Ramachandran et al. (2019), the proposed module attends to the entire input rather than a small local neighborhood. Different from Wang et al. (2020), the proposed GSA module uses a non-axial global content attention mechanism that attends to the entire image at once rather than just a row or column.

- **GSA network:** We introduce new standalone global attention-based networks that use GSA modules instead of spatial convolutions to model pixel interactions. This is one of the first works (Wang et al. (2020) being the only other work) to explore standalone global attention-based networks for image understanding tasks. Existing global attention-based works insert their attention modules into CNNs as auxiliary blocks at later stages of the network, and existing standalone attention-based networks use local attention modules.

- **Experiments:** We show that the proposed GSA networks outperform the corresponding CNNs significantly on the CIFAR-100 and ImageNet datasets while using less parameters and computations. We also show that the GSA networks outperform various existing attention-based networks including the latest standalone global attention-based network of Wang et al. (2020) on the ImageNet dataset.

## 2    RELATED WORKS

### 2.1    AUXILIARY VISUAL ATTENTION

Wang et al. (2018) proposed the non-local block, which is the first adaptation of the dot-product attention mechanism for long-range dependency modeling in computer vision. They empirically verified its effectiveness on video classification and object detection. Follow-up works extended it to

different tasks such as generative adversarial image modeling (Zhang et al., 2019; Brock et al., 2019), video person re-identification (Liao et al., 2018), image de-raining (Li et al., 2018) etc. Several recent works focused on mitigating the high computational cost of Wang et al. (2018). Chen et al. (2018); Shen et al. (2018) utilized the associative property of matrix multiplication to reduce the complexity from quadratic to linear. Huang et al. (2019) proposed to decompose global attention into row attention and column attention to save resources.

Recently, a series of works (Sun et al., 2019; Carion et al., 2020) have used Transformers (Vaswani et al., 2017) for various computer vision applications. These works first use a deep CNN to extract semantic features, and then use a Transformer to model interactions among the high-level semantic features. For example, Carion et al. (2020) used a Transformer to model object-level interactions for object detection, and Sun et al. (2019) used a Transformer to model inter-frame dependencies for video representation learning.

All these methods use attention modules as auxiliary modules to enhance long-range dependency modeling of a CNN, and relegate most of the feature extraction work to the convolution operation. In contrast, a GSA network uses attention as the primitive operation instead of spatial convolution.

## 2.2 BACKBONE VISUAL ATTENTION

Bello et al. (2019) were the first to test attention as a primitive operation for computer vision tasks. However, they used the costly non-local block (Wang et al., 2018) which prevented them from fully replacing convolutional layers. Ramachandran et al. (2019), Hu et al. (2019) and Zhao et al. (2020) solved this problem by limiting the receptive field of attention to a local neighborhood. In contrast to these works, the proposed GSA network uses global attention throughout the network and is still efficient. Recently, Wang et al. (2020) used axial decomposition to make global attention efficient. Different from them, the proposed GSA network uses a non-axial global content attention mechanism which is better than axial mechanism as later shown in the experiments.

## 3 GLOBAL SELF-ATTENTION NETWORK

### 3.1 GLOBAL SELF-ATTENTION MODULE

Let $\boldsymbol{F}^i \in \mathbb{R}^{WH \times d_{\text{in}}}$ and $\boldsymbol{F}^o \in \mathbb{R}^{WH \times d_{\text{out}}}$, respectively, denote the (spatially) flattened input and output feature maps of the proposed GSA module. Here, $W, H$ represent the spatial dimensions, and $d_{in}, d_{out}$ represent the channel dimensions. Each pixel in the output feature map is generated by aggregating information from every pixel in the input feature map based on their content and spatial positions. Let $\boldsymbol{K} = [k_{ij}] \in \mathbb{R}^{WH \times d_k}$, $\boldsymbol{Q} = [q_{ij}] \in \mathbb{R}^{WH \times d_k}$, and $\boldsymbol{V} = [v_{ij}] \in \mathbb{R}^{WH \times d_{\text{out}}}$ respectively denote the matrices of keys, queries, and values generated using three $1 \times 1$ convolutions on the input feature map $\boldsymbol{F}^i$. Here, $d_k$ denotes the number of channels used for keys and queries. Each row in these matrices corresponds to one input pixel. The proposed GSA module (see Fig. 1) consists of two parallel layers: a content attention layer and a positional attention layer.

### 3.1.1 CONTENT ATTENTION LAYER

This layer uses the keys, queries, and values to generate new features $\boldsymbol{F}^c = [f_{ij}^c] \in \mathbb{R}^{WH \times d_{\text{out}}}$ using the following content-based global attention operation:

$$\boldsymbol{F}^c = \boldsymbol{Q}\left(\rho\left(\boldsymbol{K}^\top\right)\boldsymbol{V}\right), \tag{1}$$

where $\boldsymbol{K}^\top$ denotes the matrix transpose of $\boldsymbol{K}$, and $\rho$ denotes the operation of applying softmax normalization for each row separately. This attention operation can be interpreted as first aggregating the pixel features in $\boldsymbol{V}$ into $d_k$ global context vectors using the weights in $\rho\left(\boldsymbol{K}^\top\right)$, and then redistributing the global context vectors back to individual pixels using the weights in $\boldsymbol{Q}$. The computational and memory complexities of this operation are $O(N)$ in the number of pixels.

This attention operation is similar to the attention operation used in Chen et al. (2018); Shen et al. (2018) except that it does not use softmax normalization on queries. Normalizing the queries constrains the output features to be convex combinations of the global context vectors. As these constraints could restrict the expressive power of the attention mechanism, we remove the softmax

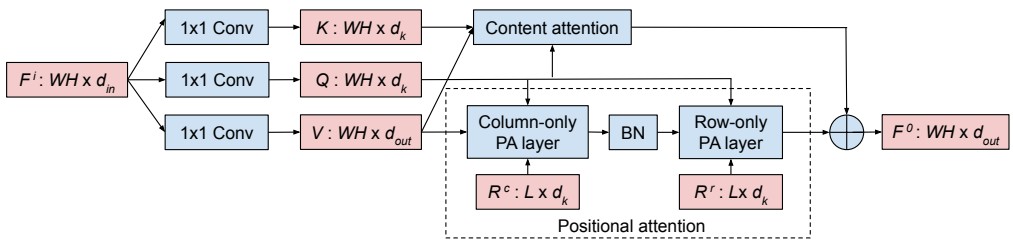

Figure 1: Proposed GSA module: The keys, queries, and values (generated using $1 \times 1$ convolutions) are processed by the content attention and positional attention layers in parallel. The positional attention layer is split into column-only and row-only positional attention layers, which use learned relative position embeddings $\boldsymbol{R}^c$ and $\boldsymbol{R}^r$ as keys. Finally, the outputs of the content and positional attention layers are summed to generate the output of the GSA module. Here, *BN* denotes batch normalization (Ioffe & Szegedy, 2015), and *PA* stands for positional attention.

normalization on queries. This allows the output features to span the entire subspace of the $d_k$ global context vectors. When we experimented with softmax normalization on the queries, the top-1 accuracy on the ImageNet validation dataset decreased significantly (1%).

### 3.1.2 POSITIONAL ATTENTION LAYER

The content attention layer does not take the spatial positions of pixels into account, and hence, is equivariant to pixel shuffling. So, on its own, it is not best-suited for tasks that deal with spatially-structured data such as images. Inspired by Bello et al. (2019); Ramachandran et al. (2019); Shaw et al. (2018), we address this issue by using a positional attention layer that computes the attention map for a pixel based on its own content and its relative spatial positions with respect to its neighbors. For each pixel, our positional attention layer attends to its $L \times L$ spatial neighbors. Inspired by the axial formulation (Ho et al., 2019; Huang et al., 2019), we implement this attention layer as a column-only attention layer followed by a row-only attention layer. In a column-only attention layer, an output pixel only attends to the input pixels along its column, and in a row-only attention layer, an output pixel only attends to the input pixels along its row. Note that a column-only attention layer followed by a row-only attention layer effectively results in information propagation over the entire $L \times L$ neighborhood.

Let $\Delta = \{-\frac{L-1}{2}, .., 0, .., \frac{L-1}{2}\}$ be a set of $L$ offsets, and $\boldsymbol{R}^c = [r^c_\delta] \in \mathbb{R}^{L \times d_k}$ denote the matrix of $L$ learnable relative position embeddings corresponding to $L$ spatial offsets $\delta \in \Delta$ along a column. Let $\boldsymbol{V}^c_{ab} = [v_{a+\delta,b}] \in \mathbb{R}^{L \times d_{\text{out}}}$ be the matrix consisting of the values at the $L$ column neighbors of pixel $(a, b)$. Let $\boldsymbol{f}^c_{ab}$ denote the output of the column-only positional attention layer at pixel $(a, b)$. Then, our column-only positional attention mechanism, which uses the relative position embeddings $\boldsymbol{R}^c$ as keys, can be described using

$$\boldsymbol{f}^c_{ab} = \left(\boldsymbol{q}_{ab}\boldsymbol{R}^{c\top}\right)\boldsymbol{V}^c_{ab}, \tag{2}$$

where $\boldsymbol{q}_{ab}$ is the query at pixel $(a, b)$. Since each pixel only attends to $L$ column neighbors, the computational and memory complexities of this column-only positional attention layer are $O(NL)$, where $N$ is the number of pixels. Similarly, a row-only positional attention layer with $O(NL)$ computational and memory complexities can be defined using $L$ learnable relative position embeddings $\boldsymbol{R}^r = [r^r_\delta] \in \mathbb{R}^{L \times d_k}$ corresponding to the $L$ row neighbors. In the case of global axial attention, the neighborhood spans the entire column or row resulting in $O(N\sqrt{N})$ computational and memory complexities.

The final output feature map of the GSA module is the sum of the outputs of the content and positional attention layers.

### 3.2 GSA NETWORKS

A GSA network is a deep network that uses GSA modules instead of spatial convolutions to model pixel interactions. Table 1 shows how a GSA network differs from various recent attention-based

Table 1: Properties of recent attention-based networks

| Method | Attention module | | | Attention + CNN combination |
|---|---|---|---|---|
| | Global | Content | Positional | |
| Wang et al. (2018) | ✓ | ✓ | | |
| Chen et al. (2018) | ✓ | ✓ | | Few attention modules are inserted in |
| Yue et al. (2018) | ✓ | ✓ | | between residual blocks |
| Shen et al. (2018) | ✓ | ✓ | | |
| Huang et al. (2019) | ✓ | ✓ | | |
| Carion et al. (2020) | ✓ | ✓ | ✓ | Attention modules are added at the end |
| Sun et al. (2019) | ✓ | ✓ | ✓ | |
| Bello et al. (2019) | ✓ | ✓ | ✓ | Convolution layers are augmented with attention modules in parallel |
| Hu et al. (2019) | | ✓ | ✓ | |
| Ramachandran et al. (2019) | | ✓ | ✓ | Convolution layers are replaced by |
| Zhao et al. (2020) | | ✓ | ✓ | attention modules |
| Wang et al. (2020) | ✓ | ✓ | ✓ | |
| **This work** | ✓ | ✓ | ✓ | |

networks. All existing works except Wang et al. (2020) either insert their attention modules into CNNs as auxiliary blocks (Bello et al., 2019; Chen et al., 2018; Huang et al., 2019; Shen et al., 2018; Wang et al., 2018; Yue et al., 2018; Carion et al., 2020; Sun et al., 2019) at later stages of the network or constrain their attention mechanism to small local regions (Hu et al., 2019; Ramachandran et al., 2019; Zhao et al., 2020). In contrast, a GSA network replaces spatial convolution layers in a deep network with a global attention module and has the ability to model long-range pixel interactions throughout the network. While Wang et al. (2020) also introduces a global attention module as an alternative for spatial convolution, their module uses axial attention mechanism for both content and positional attention. In contrast, the proposed GSA module uses a non-axial global content attention mechanism that attends to the entire image at once rather than just a row or column.

## 3.3 JUSTIFICATIONS

The proposed GSA module uses a direct global attention operation for content attention and an axial attention mechanism for positional attention.

**Why not axial content attention?** Axial attention is a mechanism that approximates direct global attention with column-only attention followed by row-only attention. In the proposed global content attention layer, two pixels $(i, j)$ and $(p, q)$ interact directly based only on their content. In contrast, in a column-followed-by-row axial content attention layer, pixels $(i, j)$ and $(p, q)$ would interact through pixel $(p, j)$, and hence, their interaction would be undesirably controlled by the content at $(p, j)$. Therefore, the proposed direct global attention is better than axial mechanism for content attention. This is also verified by the experimental results in Table 2 which show that the proposed GSA module that uses direct global content attention is significantly better than axial attention.

**Why not direct global positional attention?** It is important to attend to pixels based on relative positions (instead of absolute positions) to maintain translation equivariance. In the case of content attention, each pixel has a unique key, and hence, we can multiply keys and values first to make the attention mechanism efficient. This is not possible in the case of positional attention since the key at a pixel varies based on its relative position with respect to the query pixel. Hence, we use axial mechanism to make positional attention efficient. While axial attention is not good for content attention (as explained above), it is suitable for positional attention. The relative position between pixels $(i, j)$ and $(p, q)$ is strongly correlated to the relative positions between pixels $(i, j)$ and $(p, j)$, and between pixels $(p, q)$ and $(p, j)$. So, routing position-based interaction between $(i, j)$ and $(p, q)$ through $(p, j)$ works fine.

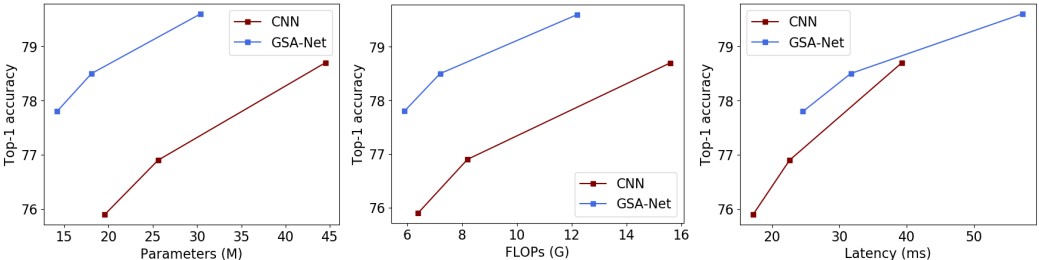

Figure 2: Comparison between ResNet-{38, 50, 101} structure-based CNNs and GSA networks. GSA networks clearly outperform CNNs while using less parameters, computations, and runtime.

## 4 EXPERIMENTS

**Model**  Unless specified otherwise, we use GSA-ResNet-50, a network obtained by replacing all $3 \times 3$ convolution layers in ResNet-50 (He et al., 2016) with the proposed GSA module. We use an input size of $224 \times 224$, and for reducing the spatial dimensions, we use $2 \times 2$ average pooling layers (with stride 2) immediately after the first GSA module in the second, third and fourth residual groups. The number of channels for $K, Q, V$ in each GSA module are set to be the same as the corresponding input features. We use a multi-head attention mechanism (Ramachandran et al., 2019; Vaswani et al., 2017) with 8 heads in each GSA module. The relative position embeddings are shared across all heads within a module, but not across modules. All $1 \times 1$ convolutions and GSA modules are followed by batch normalization (Ioffe & Szegedy, 2015).

**Training and evaluation**  All models are trained and evaluated on the training and validation sets of the ImageNet dataset (Russakovsky et al., 2015), respectively. They are trained from scratch for 90 epochs using stochastic gradient descent with momentum of 0.9, cosine learning rate schedule with base learning rate of 0.1, weight decay of $10^{-4}$, and mini-batch size of 2048. We use standard data augmentations such as random cropping and horizontal flipping. Following recent attention-based works (Ramachandran et al., 2019; Zhao et al., 2020; Wang et al., 2020), we also use label smoothing regularization with coefficient 0.1. For evaluation, we use a single $224 \times 224$ center crop. While computing FLOPs, multiplications and additions are counted separately. For reporting runtime, we measure inference time for a single image on a TPUv3 accelerator.

### 4.1 COMPARISON WITH THE CONVOLUTION OPERATION

Figure 2 compares ResNet-{38,50,101} structure-based CNNs and GSA networks. The GSA networks outperform CNNs significantly while using less parameters, computations and runtime. These results clearly shows the superiority of the proposed global attention module over the widely-used convolution operation. With increasing popularity of attention-based models, we hope that hardware accelerators will be further optimized for attention-based operations and GSA networks will become much more faster than CNNs in the near future.

### 4.2 COMPARISON WITH AXIAL ATTENTION

The GSA module uses a global content attention mechanism that attends to the entire image at once. To validate the superiority of this attention mechanism over axial attention, in Table 2, we compare the proposed GSA module with a global attention module that attends based on both content and positions similar to Ramachandran et al. (2019) but in an axial fashion. The GSA module clearly outperforms the axial alternative. Also, the performance of our axial positional attention alone is comparable to the axial attention that uses both content and positions suggesting that axial mechanism is not able to take advantage of content-only interactions (see Section 3.3 for justification).

Table 2: Comparison of the proposed GSA module with axial attention

| Attention module | Top-1 acc. | Top-5 acc. | Parameters | FLOPs | Runtime |
|---|---|---|---|---|---|
| Proposed GSA module | **78.5** | 93.9 | 18.1 M | 7.2 G | 31.7 ms |
| Axial content and positional attention | 77.5 | 93.6 | 18.1 M | 7.3 G | 32.5 ms |
| Only axial positional attention | 77.4 | 93.5 | 16.8 M | 6.4 G | 28.2 ms |

Table 3: Comparison of GSA networks ($3 \times 3$ convolutions replaced with GSA modules) with recent attention-based approaches. Note that *Wang et al. (2020)* is the conv-stem version since GSA-Net uses a conv stem. *M-ResNet-50* is a modified ResNet-50 which halves the number of input and output channels of all residual blocks and scales the number of filters in every layer by 1.125

| Structure | Method | Top-1 acc. | Top-5 acc. | Params | FLOPs | Runtime |
|---|---|---|---|---|---|---|
| ResNet-50 | Chen et al. (2018) | 77.0 | 93.5 | - | - | - |
| | Shen et al. (2018) | 77.3 | 93.6 | 26.2 M | 9.9 G | - |
| | Hu et al. (2019) | 77.3 | 93.6 | 23.3 M | 8.6 G | - |
| | Ramachandran et al. (2019) | 77.6 | - | 18.0 M | 7.0 G | 131.9 ms |
| | Yue et al. (2018) | 77.7 | 93.6 | | - | - |
| | Bello et al. (2019) | 77.7 | 93.8 | 25.8 M | 8.3 G | 34.9 ms |
| | Zhao et al. (2020) | 78.2 | **93.9** | 20.5 M | 6.6 G | - |
| | **This work** | **78.5** | **93.9** | 18.1 M | 7.2 G | 31.7 ms |
| ResNet-101 | Hu et al. (2019) | 78.5 | 94.3 | 42.0 M | 16.0 G | - |
| | Bello et al. (2019) | 78.7 | 94.4 | 45.4 M | 16.1 G | 61.9 ms |
| | **This work** | **79.6** | **94.5** | 30.4 M | 12.2 G | 57.2 ms |
| M-ResNet-50 | Wang et al. (2020) | 77.5 | - | 12.4 M | 5.6 G | 41.4 ms |
| | **This work** | **78.2** | **93.9** | 12.7 M | 6.0 G | 31.1 ms |

## 4.3 COMPARISON WITH EXISTING ATTENTION-BASED APPROACHES

Table 3 compares GSA networks with recent attention-based networks. The GSA networks achieve better performance than existing global and local attention-based networks while using similar or less number of parameters and FLOPs, except when compared to Zhao et al. (2020); Wang et al. (2020) which use slightly fewer FLOPs. Compared to local attention-based works (Hu et al., 2019; Ramachandran et al., 2019; Zhao et al., 2020), the proposed GSA network takes advantage of global attention throughout the network and produces better results. Compared to Shen et al. (2018); Yue et al. (2018); Bello et al. (2019); Chen et al. (2018) which insert a few attention modules as auxiliary blocks into a CNN, the proposed GSA network uses global attention through out the network. Compared to Wang et al. (2020), the proposed GSA network uses a non-axial global content attention which is better than axial mechanism. To report runtime for other methods, we measure single image inference time on a TPUv3 accelerator using the code provided by the corresponding authors.

## 4.4 ABLATION STUDIES

### 4.4.1 IMPORTANCE OF INDIVIDUAL COMPONENTS

As described in Section 3, a GSA module consists of three components: a content attention layer, a column-only positional attention layer, and a row-only positional attention layer. Table 4 shows the results for different variants of the proposed GSA module obtained by removing one or more of its components. As expected, the module with all three components performs the best and the content-only attention performs poorly (7.7% drop in the top-1 accuracy) since it treats the entire image as a bag of pixels. This clearly shows the need for positional attention that is missing in many existing global attention-based works (Chen et al., 2018; Wang et al., 2018; Yue et al., 2018). Interestingly, for positional attention, column-only attention performs better than row-only attention

Table 4: Comparison of different variants of the proposed GSA module

| Attention component | | | Accuracy | |
|---|---|---|---|---|
| Content | Col. | Row | Top-1 | Top-5 |
| ✓ | ✓ | ✓ | **78.5** | **93.9** |
| | ✓ | ✓ | 77.4 | 93.5 |
| ✓ | ✓ | | 76.2 | 92.7 |
| ✓ | | ✓ | 75.4 | 92.2 |
| | ✓ | | 72.6 | 90.8 |
| | | ✓ | 70.2 | 89.4 |
| ✓ | | | 70.8 | 89.5 |

Table 5: Effect of replacing convolutions with GSA modules (indicated using ✓) at different stages of ResNet-50

| Residual group | | | | Accuracy | | |
|---|---|---|---|---|---|---|
| 1 | 2 | 3 | 4 | Top-1 | Top-5 | Runtime |
| ✓ | ✓ | ✓ | ✓ | 78.5 | 93.9 | 31.7 ms |
| | ✓ | ✓ | ✓ | **78.7** | **94.1** | 28.5 ms |
| | | ✓ | ✓ | 78.5 | **94.1** | 24.5 ms |
| | | | ✓ | 77.7 | 93.7 | 23.2 ms |
| ResNet-50 CNN | | | | 76.9 | 93.5 | 22.6 ms |
| ResNet-101 CNN | | | | 78.7 | 94.4 | 39.3 ms |

(row3 vs row4 and row5 vs row6) suggesting that modeling pixel interactions along the vertical dimension is more important than the horizontal dimension for categories in the ImageNet dataset.

### 4.4.2 WHERE IS GLOBAL ATTENTION MOST HELPFUL?

Our default GSA-ResNet-50 replaces spatial convolution with the proposed global attention module in all residual groups of ResNet-50. Table 5 shows how the performance varies when global attention replaces spatial convolution only in certain residual groups. Starting from the last residual group, as we move towards the earlier stages of the network, replacing convolution with attention improves the performance consistently until the second residual group. Replacing convolutions in the first residual group results in a slight drop in the performance. These results show that the global attention mechanism is helpful throughout the network except in the first few layers. This is an expected behavior since the first few layers of a deep network typically focus on learning low-level features. It is worth noting that by replacing convolutions with the proposed GSA modules in the second, third and fourth residual blocks of ResNet-50, we are able to achieve same top-1 accuracy as convolution-based ResNet-101 while being significantly faster.

### 4.5 RESULTS ON CIFAR-100 (KRIZHEVSKY & HINTON, 2009)

Similar to the ImageNet dataset, the proposed GSA networks outperform the corresponding CNNs significantly on the CIFAR-100 dataset while using less parameters, computations, and runtime. Improvements in the top-1 accuracy with ResNet-{38, 50, 101} structures are 2.5%, 2.7% and 1.6%, respectively. Please refer to Fig. 3 and Table 6 in the Appendix for further details.

## 5 CONCLUSIONS

In this work, we introduced a new global self-attention module that takes both the content and spatial locations of the pixels into account. This module consists of parallel content and positional attention branches, whose outputs are summed at the end. While the content branch attends to all the pixels jointly using an efficient global attention mechanism, the positional attention branch follows axial formulation and performs column-only attention followed by row-only attention. Overall, the proposed GSA module is efficient enough to be the backbone component of a deep network. Based on the proposed GSA module, we introduced GSA networks that use GSA modules instead of spatial convolutions. Due to the global extent of the proposed GSA module, these networks have the ability to model long-range pixel interactions throughout the network. We conducted experiments on the CIFAR-100 and ImageNet datasets, and showed that GSA networks clearly outperform their convolution-based counterparts while using less parameters and computations. We also showed that GSA networks outperform various recent local and global attention-based networks. In the near future, we plan to extend this work to other computer vision tasks.

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

## A CIFAR-100 EXPERIMENTS

All the models are trained and evaluated on the training and test splits of CIFAR-100, respectively. They are trained for 10K steps starting from ImageNet pretrained weights using stochastic gradient descent with momentum of 0.9, weight decay of $10^{-4}$, and mini-batch size of 128. We use an initial learning rate of $5 \times 10^{-3}$ and reduce it by a factor of 10 after every 3K steps. For both training and evaluation, we use $224 \times 224$ input images.

Fig. 3 compares ResNet-{38,50,101} structure-based CNNs and GSA networks on the CIFAR-100 dataset. Similar to ImageNet results, GSA networks outperform CNNs significantly on the CIFAR-100 dataset while using less parameters, computations, and runtime. Table 6 reports all the numbers corresponding to the plots in Fig. 2 and Fig. 3.

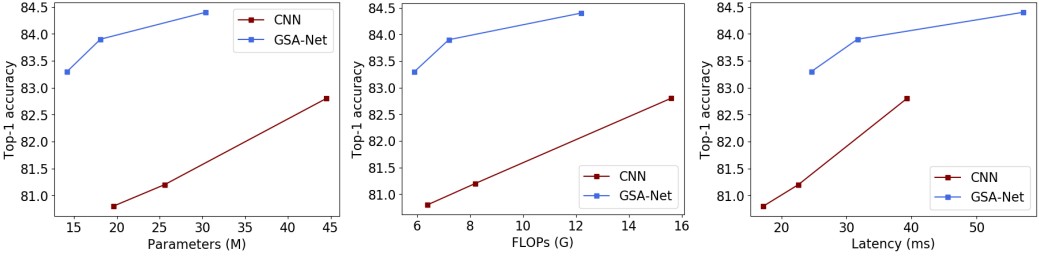

Figure 3: Comparison between ResNet-{38, 50, 101} structure-based CNNs and GSA networks. GSA networks clearly outperform CNNs while using less parameters, computations, and runtime.

Table 6: Comparison between CNNs and GSA networks

| | | ImageNet | | CIFAR-100 | | | |
|---|---|---|---|---|---|---|---|
| Structure | Operation | Top-1 acc. | Top-5 acc. | Top-1 acc. | Params | FLOPs | Runtime |
| ResNet-38 | Convolution | 75.9 | 92.9 | 80.8 | 19.6 M | 6.4 G | 17.2 ms |
| | GSA | (+1.9) **77.8** | **93.6** | (+2.5) **83.3** | 14.2 M | 5.9 G | 24.7 ms |
| ResNet-50 | Convolution | 76.9 | 93.5 | 81.2 | 25.6 M | 8.2 G | 22.6 ms |
| | GSA | (+1.6) **78.5** | **93.9** | (+2.7) **83.9** | 18.1 M | 7.2 G | 31.7 ms |
| ResNet-101 | Convolution | 78.7 | 94.4 | 82.8 | 44.5 M | 15.6 G | 39.3 ms |
| | GSA | (+0.9) **79.6** | **94.5** | (+1.6) **84.4** | 30.4 M | 12.2 G | 57.2 ms |

## B    MATHEMATICAL IMPLEMENTATION DETAILS

This section presents mathematical implementation details of the Global Self-Attention (GSA) module to supplement the high-level description in Section 3 of the paper.

For conciseness and better resemblance of the actual implementation, this section uses the Einstein notation[1]. Note that both TensorFlow Abadi et al. (2015) and PyTorch Paszke et al. (2019) provide direct support for the Einstein notation, through `tf.einsum()` and `torch.einsum()`, respectively. Therefore, there are direct TensorFlow/PyTorch transcriptions for all equations in this section.

Assume the input $X$ is a rank-3 tensor of shape $h \times w \times d$, for $h$ the height, $w$ the width, and $d$ the number of channels.

**KQV layer**    The first step is to compute the keys $K$, queries $Q$, and values $V$ from $X$ using 3 separate $1 \times 1$ (i.e. point-wise) convolution layers. Then, the module splits $K, Q, V$ each into $n$ equal-size slices along the channel dimension for the $n$ attention heads. An efficient implementation fuses the two steps into

$$K_{xynk} = W^{(K)}_{dnk} X_{xyd},$$
$$Q_{xynk} = W^{(Q)}_{dnk} X_{xyd}, \tag{3}$$
$$V_{xynv} = W^{(V)}_{dnv} X_{xyd},$$

where $W^{(K)}, W^{(Q)}, W^{(V)}$ are the corresponding weights, $x, y$ are the spatial dimensions, $n$ is the head dimension, and $d, k, v$ are the channels dimensions for the input, the keys and queries, and the values, respectively.

**Content attention**    As Section 3 of the paper describes, within each head the module uses matrix multiplication to implement content attention. The actual implementation parallelizes the process across all heads by computing

$$\hat{K} = \sigma(K),$$
$$C_{nkv} = \hat{K}_{xynk} V_{xynv}, \tag{4}$$
$$Y^C_{xynv} = Q_{xynk} C_{nkv},$$

where $\sigma$ represents softmax along the spatial dimensions $(x, y)$.

**Positional attention**    The positional attention layer consists of a column-only attention sub-layer, a batch normalization layer, and a row-only attention sub-layer. Since the column-only and row-only sub-layers are symmetric, this section only presents the implementation for the column-only sub-layer.

---

[1]The Einstein notation is a compact convention for linear algebra operations Albert Einstein developed. https://en.wikipedia.org/wiki/Einstein_notation provides a reference. https://ajcr.net/Basic-guide-to-einsum/ gives an intuitive tutorial.

The layer maintains a relative position embedding matrix $R \in \mathbb{R}^{(2h-1) \times k}$, for $h$ the image height and $k$ the number of channels. Each of the $2h - 1$ rows corresponds to a possible vertical relative shift, from $-(h - 1)$ to $h - 1$. The first step is to re-index this matrix from using relative shifts to absolute shifts. To achieve this goal, the module creates a re-indexing tensor $I$ where

$$
\begin{aligned}
I_{x,i,r} &= 1, \quad \text{if } i - x = r \ \& \ |i - x| \leq L, \\
I_{x,i,r} &= 0, \quad \text{otherwise},
\end{aligned}
\tag{5}
$$

where $L$ is the maximum relative shift to attend to. The default version of GSA sets $L = max\{h, w\}$ so that the positional attention is global.

Then, the module computes the position embedding tensor whose indices are the absolute shifts as

$$
P_{xik} = I_{xir} R_{rk}.
\tag{6}
$$

Now, the output of the column-only attention sub-layer is

$$
\begin{aligned}
S_{xyin} &= Q_{xynk} P_{xik}, \\
Y^H_{xynv} &= S_{xyin} V_{iynv}.
\end{aligned}
\tag{7}
$$

After obtaining $Y^H$, the module applies batch normalization to it and uses it as the input to the row-only sub-layer to generate $Y^W$ as the final output of the positional attention layer.

**Final fusion** After computing the outputs of the content and positional attention layers, the final output is simply

$$
Y = Y^C + Y^W.
\tag{8}
$$

**Comparison to competing approaches** The implementation of the GSA module only consists of 8 Einstein-notation equations and 5 other equations, each of which corresponds to one line of code in TensorFlow or PyTorch. The implementation is substantially simpler in comparison to competing approaches Ramachandran et al. (2019); Zhao et al. (2020) using local attention which requires custom kernels.

