# OpenReview forum: "Global Self-Attention Networks for Image Recognition"
_ICLR.cc/2021/Conference — Reject_

### Official Review · AnonReviewer4 · 2020-10-27
**Good but needs more thorough experimental evaluation**

**Rating:** 5
**Confidence:** 5

**Review:**

There have been multiple attempts to use self-attention in computer vision backbones for image classification and object detection. Most of these approaches either tried to combine convolution with global self-attention, or replace it completely with local self-attention operation. The proposed approach naturally combines the two, by employing query-key-value switching trick, with axial positional attention.

The approach is very similar to Axial Deeplab (Wang et al. 2020, ECCV 2020 Spotlight), with the main difference that Axial Deeplab uses axial self-attention in the content part, instead of global self-attention, for computational efficiency reasons. The authors claim superiority of their approach, and provide an experimental comparison, showing that there is almost no benefit from axial content attention, which is suspicious - could be an implementation issue?

Overall, the benefit of content attention is not clear from the experiments, as it has more parameters, it is not surprising that it yields higher accuracy and slower runtime. An experiment with the same number of parameters with/without content part is missing.

Regarding softmax on queries, I think the motivation behind was activation normalization. I am curious if it poses additional training difficulties and if the authors tried other activations similar to Katharopoulos et al. (Transformers are RNNs), e.g. ReLU and scalar normalization.

CIFAR-100 experimental setup is uncommon and very surprising, as the authors use ImageNet pretrained models and finetune on CIFAR-100 with large input images, in contrast to training from scratch on CIFAR-100. I suggest that these experiments are updated, otherwise it is extremely difficult to reproduce the paper with a limited computational budget.

Regarding the reproducibility, releasing the code for such methods must be a requirement, as reproducing similar approaches like stand-alone self attention is notoriously difficult, and I don’t think the community managed to do it.

Axial Deeplab paper has been online for 6 months before ICLR submission deadline, the novelty over it is very limited, but I would leave for ACs to decide if it is concurrent or prior work. Disregarding this, the paper has valuable findings but needs more rigorous experimental evalution, as I suggested in my review.

---

> ### Author Response · Authors · 2020-11-17
> **Response to Reviewer 4**
>
> # Correctness of comparison with Axial-ResNet
>
> Speed comparison:
>
> - In our content branch, we use the $\boldsymbol{Q}\left(\boldsymbol{K}^\mathsf{T}\boldsymbol{V}\right)$ global attention, similar to double attention (Chen et al., 2018) and efficient attention (Shen et al., 2018). Perhaps surprisingly this type of global self-attention is faster than axial attention. The reason is that $\boldsymbol{Q}\left(\boldsymbol{K}^\mathsf{T}\boldsymbol{V}\right)$ attention's complexity is $O(n)$, since it does not generate any intermediate features larger than $n \times d$, for $d$ the respective channel depth. In contrast, axial attention has complexity $O\left(n\sqrt{n}\right)$, since for every of the $n$ pixels, it generates an attention map of size $\sqrt{n}$ (i.e. height or width).
> - Our runtime comparison is accurate and correct since we are benchmarking the code from the authors of Axial-ResNet (Wang et al., 2020) on the exact same setup as our GSA-Net.
>
> Accuracy comparison:
>
> - We carefully checked our code and do not believe that there is an implementation issue. An additional piece of evidence is that our implementation (Table 2) has the accuracy as the original author's implementation (Table 3), and we checked with them that our training setups are highly similar.
>
> # Ablation studies for no-content/no-position versions while matching parameter count
>
> We will do these experiments and reply here when they finish.
>
> # ReLU or scalar normalization
>
> We will do these experiments and reply here when they finish.
>
> # Setup for CIFAR-100 experiments
>
> Thank you for pointing out. We will find a typical setup from the literature, redo the experiments, and reply here when they finish. We will be grateful if you can give any pointer.
>
> # Reproducibility
>
> We have applied for anonymous open-source permission from our company. We hope the permission can be granted in time. If so, we will update the supplementary materials to include anonymized code.
>
> # Concurrency with Axial-DeepLab
>
> We acknowledge that it has been several months since the publication of Axial-DeepLab. However, the ideation and the majority of the experiments were done before Axial-DeepLab appeared on arXiv. Therefore, we believe GSA-Net is concurrent with Axial-DeepLab from our perspective.

---

> > ### Author Response · Authors · 2020-11-25
> > **Response to remaining questions**
> >
> > We sincerely apologize that we did not manage to conduct experiments we promised in the previous reply, including experiments for no-content/no-position version with matching parameter counts and ReLU/scalar normalization and CIFAR-100 experiments with more typical setups, in the duration of the discussion phase.

---

> ### Author Response · Authors · 2020-11-25
> **Code submission**
>
> We are glad that our code has gone through internal review for anonymous submission. We have uploaded a ZIP as supplementary materials, which contains our model code, including definitions of the model and all layers and operators. It does not currently contain the training and data pipelines since there are dependencies on internal infrastructure that we haven't fully cleaned.
>
> For the data pipeline, we use the standard ImageNet dataset and simple augmentations (random resizing, random cropping, and random flipping). For the training pipeline, we have described the set up in the paper in detail.
>
> We are continuing working on open-sourcing our full code base. You can expect it in a few weeks.

---

### Official Review · AnonReviewer2 · 2020-10-28
**A attention-based model that rivals convolutional models on ImageNet**

**Rating:** 4
**Confidence:** 4

**Review:**




**Update after rebuttal** : I want to thank the authors for their rebuttal. However, after reading all responses and all new results presented, I still think that most of the weaknesses of this paper are still present. Two notes that I urge authors to take into account:

* Even the updated Figs 2 and 3 are highly misleading: sure, the gains over a baseline ResNet network are big, but the comparisson should be vs other backbones that combine convolutions with attentions.
* The authors should clearly say that they do use a convolution for conv1, and their modules start on top of that first spatial convolution.

----------------------------------------

Summary:
The authors present a composite attention module that complement the global attention with axial attentions over rows and columns. They replace 3x3 convolutions with this module on ResNets and show they can outperform convoultion and convolution+ attention based models on ImageNet.

Strong points: The authors present a method that replaces almost all convolutions of a CNN with self-attention variants and gives high results for ImageNet.

Weak points:
Many parameters of the method are chosen arbitrary and are not really ablated; the same stands for alterations in eg the "content attention" layer which is a variant of self-attention, but not exactly the same, since the softmax is applied only on the key values. For a method that argues that one should replace convolutions with attentions, a lot more ablations on hyperparameter stability and effect would be needed, not just a "winning combination", as well as experiment on more than one dataset (CIFAR experiments are for transfer learning).

Specifically:

A) In general, a lot of ablations are missing. The effect of the number of heads, the dimension parameter $d_k$ and the very important region parameter $L$ are not ablated. In fact, parameter $L$ is never explicitly set in the main text - the end of the appendix states "the default version of GSA sets L = max{h, w}" - this should be clearly set and ablated in the main text.  The term "global axial attention" is mentioned in the text, but the text does not say that global axial attention is the "default case".

B) Sure, not having a softmax as in the common decomposed self-attention allows for more flexibility, but the gains from this change are not really isolated nor ablated; In the text the authors say that there is a decrease by 0.9% on ImageNet without specifying any details (which model? compared to what?). This seems like a tweak that affect performance significantly, that is not ablated.
Qb1) What is the performance of the proposed module when keeping everything the same but just substituting the proposed "content attention" with the common way of using softmax, eg what is used in (Chen et al 2018) or (Shen et al 2018)? If there is a large drop, how would you explain this?

C) The section that presents the GSA network (3.2) is very short, incomplete and  highly unclear:
Qc1)  the authors say they "replace all 3 × 3 convolution layers in ResNet-50" , but the first conv1 in ResNet is not 3x3: Do you replace *all* convolutions? even the very first one, or is there a 7x7 convolution in your GSA network? If so, this should be clearly stated and not implied.
Qc2) How many layers of self-attention do you have in each block, is it exactly as many as convolutions?  how do you end up with 18M Parameters and these FLOPS?

D) The paper presents results *only* on the ImageNet dataset (There are some results on CIFAR in the appendix, but models there are only fine-tuned from ImageNet pretrained, and not from scratch); This is a dataset that has no true test set and all methods are overfitting the validation set; this is a dataset where different hardware setups give different baseline numbers. I understand that this is common practice, but, as most other papers do, I would also expect results at at least one more dataset or task to be fully convinced that this approach has something novel to offer. What is more, the curves in Figures 2,3 are against the vanilla ResNet architecture and seem a bit outdated;  a number of more recent papers have taken convolutions to much higher performance with minimal overheads (eg SE-Nets) that would not really show such big differences.
Qd1) Baseline performance (on the same setup) is highly important and not really presented in Table 3: what would a baseline ResNet-50 achieve at the authors's hardware, and for the same training setup (90 epochs using stochastic gradient descent with momentum of 0.9, cosine learning rate schedule with base learning rate of 0.1, weight decay of 10−4, and mini-batch size of 2048 as well as with the same label smoothing)? This would enable a more fair comparison between Convolutions and a network of (almost) only self-attentions.
Qd2) Are Resnet numbers in Figures 2,3 reproduced or copied from the 2016 resnet paper?

E) In a missing related work [Chen et al "Graph-based global reasoning networks." CVPR 2019] use global attention + a graph convolution in a module they call GloRe; the graph convolution there seems to be something similar to axial attention, but over nodes on a graph and not positions. Similar to this paper that restricts softmax only on the keys , the softmax in GloRe  is also removed for similar reasons. This is a paper that would be nice to be discussed and compared to - a comparison is also missing from Table 3; that paper reports that a Resnet-50 with 3 Glore units reaches 78.4 top-1 accuracy with 5.2 GFLOPS (and 30M params), vs 78.5 for the approach proposed here with 7.2 GFLOPS.

Notes and questions:
* The name of the module focuses on "global" but this is a characteristic of many other attentions cited (Wang et al. (2018); Yue et al. (2018); Chen et al. (2018); Shen et al. (2018); Huang et al. (2019)) and doesn't seem fitting to be the main thing titled here. The approach adds positional/axia attention side by side to global; this could/should be reflected in the title
* It would be interesting to see if distillation could help this model as much as it does convolutional resnets.
* Figures 2,3 would be clearer if markers were indicated for the 3 datapoints of each line.

---

> ### Author Response · Authors · 2020-11-17
> **Response to Reviewer 2**
>
> Thank you for your time assessing our paper and your valuable feedback.
>
> # Additional ablation studies
>
> Number of attention heads:
>
> - Ablations studies are underway. We will reply here once they finish.
>
> Dimensionality of keys:
>
> - Ablations studies are underway. We will reply here once they finish.
>
> Spatial extent $L$:
>
> - Sorry for our unclear presentation. We always set $L = max(h, w)$ so that the position branch is also global. We consider globality to be our key differentiation versus competitors. Therefore, the paper did not have ablation studies on that. We will update our paper to remove the notion of $L$ to improve clarity.
>
> # Ablation on attention normalization (softmax)
>
> Ablations studies are underway. We will reply here once they finish.
>
> # Clarification on network architecture
>
> We only replace 3x3 convolutions. I.e., we don't replace the single 7x7 convolution in the beginning, or the 1x1 convolutions (which in a sense are not convolutions). We will update the paper to clarify on this setting.
>
> # Additional validation dataset and comparison against ResNet
>
> Additional dataset:
>
> - We apologize the the time frame of the discussion phase doesn't allow us to get results on additional datasets. However, we plan to update the CIFAR-100 results using a setup that is much more common and comparable to other works in the literature.
>
> Version of ResNet to compare against:
>
> - We compared against the vanilla ResNet because our GSA-Net is also vanilla, free of most performance tricks. Our ResNets and GSA-Nets have the exact same setup for architecture, training (hardware, batch size, loss, optimizer, training length, etc.), and testing. The only difference is that we swap the 3x3 convolutions in the middle of every bottleneck ResBlock out for a single GSA-Net.
> - We plan to present an additional set of experiments comparing SE-ResNet and SE-GSA-Net, to showcase that performance tricks for ResNet transfer to GSA-Net.
>
> Source of the numbers for ResNet:
>
> - We reproduce ResNet on our own setup, which is exactly the same as our setup for GSA-Net (architecture, training, and testing).
>
> # Comparison with GloRe
>
> GloRe is similar to double attention (Chen et al., 2018) and efficient attention (Shen et al., 2018). All three works proposed some efficient variant of attention as an *auxiliary* module for CNNs. GloRe differs by:
>
> 1. Having further transformations in the interaction space (i.e. the global context in attention terms);
> 2. Decisions to share or not share specific transformations (e.g. GloRe's $\boldsymbol{B}$ is effectively $\boldsymbol{Q}$ and $\boldsymbol{K}$ for attention methods).
>
> We will add relevant discussion to the paper.
>
> An **important** note on FLOPs: we count multiplications and additions separately, while GloRe counts one multiplication and one addition as ONE operation. Therefore, there is a 2x gap between the two counting standards. By GloRe's standard, we only have 3.6 GFLOPs, 30% less than GloRe.
>
> # Other suggestions
>
> Title:
>
> - Thank you for pointing out the issue. We believe our main distinction is that we proposed the first fully-global, fully-attentional backbone component for vision. Our emphasis on globality is explained in the section for "Additional ablation studies". Therefore, we propose to rename the method to *global fully-attentional networks (GFAN or gFAN)". Would you think this is a better name?
>
> Distillation:
>
> - Thank you for the suggestion. We will try to get some results before the discussion deadline.
>
> Figures 2 and 3:
>
> - Thank you for the suggestion. We will update the figure. We will reply here when we finish the revision of the paper.

---

> > ### Comment · AnonReviewer2 · 2020-11-18
> > **Response to the first author note**
> >
> > Thank you for the response and the clarifications.
> >
> > Regarding the name, 'fully-attentional networks" seems to indeed be closer ("global self-attention network" for me implied that the self attention is global - having a coma would help "global, self-attention networks" but again global and self-attention are redundant terms, since the vanilla self attention is global. "fully-attentional networks" is better, but again, there is still an important vanilla 7x7 first convolution for Conv1 that enables the computational complexity to be manageable for the first attentional layer. So it is also misleading, unless there is a clear figure, that shows the network architecture (which is indeed missing from a paper presenting a new architecture) - I would encourage the authors to add such a figure regardless of name.

---

> > > ### Author Response · Authors · 2020-11-18
> > > **Naming and architecture graph.**
> > >
> > > Thank you for the suggestion. We will add the figure.
> > >
> > > Regarding the name, do you prefer fully-attentional networks (FAN) over global fully-attentional networks (GFAN)?

---

> > ### Author Response · Authors · 2020-11-25
> > **Response to remaining questions**
> >
> > # Additional dataset
> >
> > We acknowledge that evaluation on a single dataset may not be sufficient evidence for generalizability. We apologize that we did not manage to conduct experiments on other tasks during the discussion phase.
> >
> > # Results for SE-related experiments
> >
> > We believe that SE should bring similar benefits to GSA-Nets as it does to CNNs. However, we apologize that we did not manage to finish experiments to demonstrate this claim.
> >
> > # Distillation
> >
> > It is an interesting direction to try out in the future, but we did not find time to do it in the discussion phase and felt it might not fit in the scope of this work.
> >
> > # Format of figures 2 and 3
> >
> > We updated the figures in the paper. In addition, Table 6 holds the raw data for these figures.

---

> ### Author Response · Authors · 2020-11-24
> **Ablation on the number of attention heads**
>
> | Head count | Top-1 acc. | Top-5 acc. | Parameters (M) | FLOPs (G) | Latency (ms) |
> | - | - | - | - | - | - |
> | 1 | 74.6 | 91.9 | 18.3 | 8.3 | 31.5 |
> | 2 | 76.6 | 93.0 | 18.2 | 7.6 | 30.4 |
> | 4 | 77.9 | 93.6 | 18.1 | 7.3 | 29.2 |
> | 8 | **78.5** | **93.9** | 18.1 | 7.2 | 31.7 |
> | 16 | 78.4 | **93.9** | 18.1 | 7.1 | 33.3 |
> | 32 | 78.2 | 93.7 | 18.1 | 7.1 | 36.0 |
>
> As the table shows, 8 attention heads give an optimal tradeoff between accuracy and cost. With less heads, the expressiveness of each attention layer is limited due to an inadequate number of attention patterns. In addition, the FLOPs and number of parameters get high due to wider heads. With more than 8 heads, the channel depth of each head is too small to allow high-quality modeling of interactions. Further, due to small dimensions on the intermediate tensors, accelerator utilization decreases. Consequently, the speed is slower despite fewer FLOPs.

---

> ### Author Response · Authors · 2020-11-25
> **Ablation on the dimensionalities of queries, keys, and values.**
>
> We conducted additional ablation experiments on the channel depths of queries, keys, and values. Note that queries and keys always have the same dimensionality, so we only specify `key_depth_multiplier`. We apologize that experiment `key_depth_multiplier=1.0, value_depth_multiplier=2.0` will not finish before the deadline.
>
> | Model | Key depth multiplier | Value depth multiplier | Top-1 acc. | Top-5 acc. | Parameters (M) | FLOPs (G) |
> | - | -: | -: | -: | -: | -: | -: |
> | ResNet-50 | - | - | 76.9 | 93.5 | 25.6 | 8.2 |
> | GSA-ResNet-50 | 0.5 | 0.5 | 76.9 | 93.1 | 13.6 | 5.0 |
> | | 0.5 | 1.0 | 78.1 | 93.8 | 15.8 | 6.3 |
> | | 0.5 | 2.0 | 78.9 | 94.1 | 23.1 | 9.0 |
> | | 1.0 | 0.5 | 77.0 | 93.3 | 14.9 | 5.8 |
> | | 1.0 | 1.0 | 78.5 | 93.9 | 18.1 | 7.2 |
> | | 1.0 | 2.0 | - | - | 24.4 | 9.9 |
> | | 2.0 | 0.5 | 77.3 | 93.3 | 17.5 | 7.5 |
> | | 2.0 | 1.0 | 78.6 | 93.9 | 20.6 | 8.9 |
> | | 2.0 | 2.0 | 79.1 | 94.2 | 26.9 | 11.8 |
>
> This set of experiment show that it is more efficient to allocate more channel depth to the values rather than the keys with the same parameter or computational budget. The comparison between `key_depth_multiplier=0.5, value_depth_multiplier=2.0` and `key_depth_multiplier=2.0, value_depth_multiplier=1.0` clearly shows this observation.
>
> Further, these experiments show that the effectiveness of GSA-Net is stable to the choice of these hyperparameters. All models outperform the convolutional ResNet-50, with the only exception of the smallest model (`key_depth_multiplier=0.5, value_depth_multiplier=0.5`) matching the performance of ResNet-50 with almost half the parameters and computation.

---

> ### Author Response · Authors · 2020-11-25
> **Ablation study on the spatial extent**
>
> As we stated in an earlier response, we strive for globality in this work and consequently set the spatial extent $L$ always to $2\operatorname{max}(h, w) - 1$. However, we are grateful that you pointed out the need to ablate on this parameter, which can serve to validate the importance of globality. The following table presents the results:
>
> | Spatial extent | Top-1 acc. | Top-5 acc. | Parameters (M) | FLOPs (G) | Latency (ms) |
> | -: | -: | -: | -: | -: | -: |
> | 3 | 77.8 | 93.6 | 18.0 | 7.2 | 31.2 |
> | 7 | 78.4 | 93.9 | 18.0 | 7.2 | 31.2 |
> | 15 | 78.4 | 94.1 | 18.1 | 7.2 | 31.2 |
> | 31 | 78.4 | 93.9 | 18.1 | 7.2 | 31.4 |
> | 63 | 78.5 | 94.0 | 18.1 | 7.2 | 32.2 |
> | $\infty$ | 78.5 | 93.9 | 18.1 | 7.2 | 31.3 |
>
> (Note that the maximum resolution for the input to a GSA module is $56 \times 56$. Therefore, a spatial extent of 111 is effectively global.)
>
> We can observe a steady increase of accuracy as the spatial extent grows. However, most of the benefits are obtained at a spatial extent of 7. Nonetheless, note that going from 3 all the way to $\infty$ incurs negligible costs in parameters, computation, or latency. This is because we implement local variants of GSA-Net by masking positional logits beyond the spatial extent, which does not save any computation or latency. (It marginally saves parameters since it requires fewer positional embeddings.) The difference in inference speed is mostly due to random variance.
>
> If we alternatively choose to actually implement local attention similar to SASA (Ramanchandran et al., 2020), it saves some theoretical computation but is practically substantially slower (~4x) due to the use of primitives that are highly accelerator-unfriendly.
>
> Therefore, having a global spatial extent:
> 1. is essentially free compared to more restricted variants;
> 2. gives a simpler implementation since it doesn't require masking.

---

> ### Author Response · Authors · 2020-11-25
> **Ablation on attention normalization (softmax)**
>
> We apologize for unclear presentation on this issue in the paper. The 0.9% difference was from earlier ablation studies on a setting different from the current setting in the paper. Therefore, we conducted additional ablation studies on attention normalization. The following table presents the results.
>
> | Model | Softmax on $\boldsymbol{Q}$ | Softmax on $\boldsymbol{K}$ | Top-1 acc. | Top-5 acc. | Parameters (M) | FLOPs (G) | Latency (ms) |
> | - | :-: | :- :| - :| -: | -: | -: | -: |
> | GSA-ResNet-50 | T | T | 78.0 | 93.8 | 18.1 | 7.2 | 33.1 |
> | | F | T | 78.5 | 93.9 | 18.1 | 7.2 | 31.7 |
>
> As we see, surprisingly, switching to using two softmax functions instead of a single one decreases performance by 0.5%. The intuition behind this change, as we justified in the paper, is based on the interpretation of $\boldsymbol{Q}\left(\boldsymbol{K}^\mathsf{T}\boldsymbol{V}\right)$ attention that it is generating a set of template attention maps each summarizing a global, semantic aspect of the input and letting each pixel decide coefficients to combines them. Per this interpretation,  we can relax the constraint for each $\boldsymbol{q}_i$ to sum to $1$, thus allowing non-convex combinations of the template attention maps. This change enhances the expressiveness of the module.

---

### Official Review · AnonReviewer3 · 2020-10-28

**Rating:** 5
**Confidence:** 3

**Review:**

# Post-rebuttal update

I would like to thank the authors for the detailed feedback. I am now convinced about the statistical significance of the results. Regarding the additional study, while it is true that the combination of the changes, in addition to the softmax, was what made the results improve, the change is quite minor. Also, the biggest change comes from the fact that softmax is removed. The reviewer also finds the explanation of why this happens to be handwavy.

Given the concurrent work [Wang et al. 2020] and the incremental nature of the innovation, the reviewer is not sure of the benefits of the current paper to be published at ICLR.

# Summary

The paper proposes a Global Self Attention (GSA) layer, that encapsulates both global and local self-attention, by computing them in parallel and merging the two attentions together. The method is tested on ImageNet and CIFAR-100, demonstrating state-of-the-art results.

While the paper delivers state-of-the-art performance, the novelty of the paper is weak. The enhancements in this paper are also minor, making the paper borderline. The paper currently makes strong claims about the benefit of the proposed method, which are not all obvious. Thus, my preliminary rating for the paper is borderline reject.

# Strengths

The main strength of the paper is its performance and the extensive comparison against the state-of-the-art. The method outperforms all compared methods, albeit by a small margin. CIFAR-10 results are also interesting, but there is no comparison against other self-attention.

The paper is easy to follow, with detailed coverage of related works.

The ablation study demonstrates which components are important.

# Weaknesses

## Weak novelty, sigmoid, & regarding Wang et al. (2020)

Components that constitute GSA are already shown to be highly effective. The main novelty of the paper then seems to be that the softmax is not used at certain locations and that the two different types of attention are combined together. Thus, the novelty of the paper is limited, especially considering the fact that the gain in performance is marginal. Furthermore, while the focus is on the combination of the two types of attention, which does seem to help in the ablation study, I wonder whether the removal of the softmax had a larger effect. This makes the argument of the paper weak. This leads to my second concern about the paper.

The removal of the sigmoid seems to have had a large effect. At the end of section 3.1.1, it is mentioned that removing softmax increases performance by 0.9% absolute. Considering that the amount of gain from global attention is around 1.1% absolute, this means that without the softmax, there is almost no gain from introducing global attention. Furthermore, I wonder if Wang et al. (2020), without the softmax as in this paper would provide enhanced results as in this paper.

Thus, it is my current understanding that the method, with softmax and without content attention is similar, if not identical, to Wang et al. (2020). An additional ablation study where softmax is removed from Wang et al. (2020)'s method is necessary to clarify where the performance gain is really coming from.

Also, I am not sure why the paper has chosen the Conv-stem from Wang et al. (2020) and not the full version. Could the authors clarify this?

## Not significantly better

The tone of the paper, regarding the experiments, also needs to be weakened. The method, compared to prior work, improves existing work only slightly. For example, in Table 3, the gap in top-1 performance 0.3% in the case of Resnet-50, 0.9% in ResNet-101, and 0.1% compared to the full version of Wang et al. (2020). I also wonder if this satisfies statistical significance, given that each training session will give slightly varying results.

---

> ### Author Response · Authors · 2020-11-17
> **Response to comments by Reviewer 3**
>
> Thank you for your time assessing our paper and your valuable feedback.
>
> # Novelty
>
> We would like to gently point out that GSA-Net is concurrent with Axial-DeepLab (Wang et al., 2020). We acknowledge that it has been several months since the publication of Axial-DeepLab. However, the ideation and the majority of the experiments were done before Axial-DeepLab appeared on arXiv.
>
> # Ablation on attention normalization (softmax)
>
> The number we mentioned was from an ablation study on an earlier iteration of our model. We are doing these ablation studies on our current setup now and will reply here once we have results.
>
> # Comparison with Wang et al. (2020)
>
> Ablation on the combinations of attention normalization with Axial-ResNet/GSA-Net:
>
> - The previous section answered this question.
>
> Why we compared against the conv-stem Axial-ResNet:
>
> - We chose this version of Axial-ResNet because we are using conv stem for GSA-Net.
>
> # Tone of paper
>
> We would like to argue that the improvement is significant, not in terms of absolute accuracy gain, but in terms of performance-runtime tradeoff.
>
> 1. For all our experiments in Table 3, we ran three times and took the average.
> 2. On ResNet-50, although the absolute accuracy difference is only 0.3%, we have reasons (based on our discussion with the authors of Zhao et al. (2020) and neural network acceleration engineers) to believe that the runtime of Zhao et al. (2020) is similar to Ramanchandran et al. (2020), which is 4 times slower than us. The reason is that global attention utilizes standard NN operations (e.g. matrix multiplication), while local attention needs specialized primitives. Those primitives are very hard to optimize on modern accelerators (e.g. GPUs, TPUs).
> 3. On ResNet-101, we believe a 0.9% margin from a 3-run average is considered significant on ImageNet.
> 4. Comparison with Wang et al. (2020) is covered in the previous section.

---

> ### Author Response · Authors · 2020-11-25
> **Ablation on attention normalization**
>
> We conducted additional ablation studies on attention normalization. The following table presents the results.
>
> | Model | Softmax on $\boldsymbol{Q}$ | Softmax on $\boldsymbol{K}$ | Top-1 acc. | Top-5 acc. | Parameters (M) | FLOPs (G) | Latency (ms) |
> | - | :-: | :- :| - :| -: | -: | -: | -: |
> | Axial attention | T | T | 78.0 | 93.8 | 18.1 | 7.3 | 34.2 |
> | | F | T | 77.5 | 93.6 | 18.1 | 7.3 | 32.5 |
> | GSA module| T | T | 78.0 | 93.8 | 18.1 | 7.2 | 33.1 |
> | | F | T | 78.5 | 93.9 | 18.1 | 7.2 | 31.7 |
>
> (Note that *axial attention* in this table refers to the same implementation as *axial position and content attention* in Table 2, except perhaps different normalization. The corresponding row in Table 2 uses softmax only on $\boldsymbol{K}$.)
>
> As we see, surprisingly, switching to only softmax on $\boldsymbol{K}$ decreases performance, while the same change increases performance for the GSA module. A potential explanation links to the interpretation of $\boldsymbol{Q}\left(\boldsymbol{K}^\mathsf{T}\boldsymbol{V}\right)$ attention that it is generating a set of template attention maps each summarizing a global, semantic aspect of the input and letting each pixel decide coefficients to combines them. It is following this interpretation that we relaxed the constraint for each $\boldsymbol{q}_i$ to sum to $1$. However, this interpretation does not apply to axial attention equally well. On axial attention, the attention maps summarizing global, semantic aspects will be summarizing column/row-wise, semantic aspects. A single column or row might not provide a sufficiently stable summarization for each semantic aspect. Therefore, relaxing the constraint of $\operatorname{sum}\left(\boldsymbol{q}_i\right)$ might not make sense for axial attention (rather than global attention).

---

### Official Review · AnonReviewer1 · 2020-11-02
**Not so convinced by the claims in the paper**

**Rating:** 4
**Confidence:** 5

**Review:**

Summary:
This paper proposes global self-attention networks for image recognition. The proposed GSA module is consists of two parallel layers,  a content attention layer, and a positional attention layer. Experimental evaluations are conducted on ImageNet and Cifar100, and the results are promising.

Clarity:
1. I think this paper is moderate. The content attention layer can be treated as calculating the correlations between each slice of the feature maps, then reweighting/aggregating information through the channel dimension. It is not spatial attention and maybe we cannot say it is global attention.
2. The complexity of the proposed content attention layer is O(Nd_kd_out), and classical global attention like non-local is O(NNd_k). The author should not ignore the channel values as they are large in the top layers.
3. From table 4, the position attention layer is much more important than the content attention layer (77.4 vs 70.8), the position attention is local attention (not global) and also the content is not global spatial attention (for capturing long-range context). Thus I am not convinced by the claim of the paper. Besides, from previous papers (Hu et al. (2019), Ramachandran et al. (2019), Zhao et al. (2020)), we know that local self-attention is already enough, and global attention is not really needed.

---

> ### Author Response · Authors · 2020-11-14
> **Response to comments by Reviewer 1**
>
> Thank you for your time assessing our paper and your valuable feedback.
>
> # Globality of the efficient content attention branch
>
> The attention mechanism essentially computes $\boldsymbol{Q}\boldsymbol{K}^\mathsf{T}\boldsymbol{V}$, with various normalization in between. Due to the associativity of matrix multiplication, the order of actual multiplication has no effect on the outcome when using the same normalization. Therefore, it should also not affect the interpretation. In a sense, for dot-product attention, spatial attention and channel attention are the same.
>
> The analysis up to now ignores the normalization. Now, let's look at the effect of normalization on the interpretation. In our setting, we do $\boldsymbol{Q}\operatorname{softmax}_{\text{column}}\left(\boldsymbol{K}^\mathsf{T}\right)\boldsymbol{V}$, which does not specifically point toward channel attention instead of spatial attention. From a spatial attention perspective, you can regard our content branch as:
>
> 1. computing a set of $d_k$ global template attention maps shared by all pixels (i.e. $\boldsymbol{K}$);
> 2. computing a set of $d_k$  coefficients at each pixel ($\boldsymbol{Q}$), corresponding to the $d_k$ template attention maps;
> 3. aggregating the values ($\boldsymbol{V}$) by the $d_k$ template attention maps to form $d_k$ global context vectors each describing a global, semantic aspect of the input;
> 4. distributing the global context vectors according to the local coefficients.
>
> # Complexity w.r.t. channel depth
>
> Thank you for pointing this out. We acknowledge that the complexity of conventional attention is $O(n^2d)$ and of our efficient content attention is $O(nd^2)$, assuming $d = d_k = d_\text{out}$. However, please note that:
>
> 1. We use the multi-head mechanism, with 8 heads in the default setting. This makes it that even in later layers, $d$ is still small. E.g., in a GSA-ResNet, the widest attention layer has $d = \frac{512}{8} = 64$, which still does not make a bottleneck.
> 2. Since our content attention is linear, if in extreme cases we do have $d \gg n$, we can switch to the $\left(\boldsymbol{Q}\boldsymbol{K}^\mathsf{T}\right)\boldsymbol{V}$ computational order with no loss in performance, even after training is done.
>
> # Importance of globality
>
> We would like to argue that our network is indeed global, in the sense that:
>
> 1. The content branch is global as we argued in response to Q1;
> 2. The positional branch is also global, since through a layer of full-row attention  and another of full-column attention, a pixel gets to see the entire input in this two-step manner. We apologize that introducing the $L$ parameter in the paper confuses the reader. Because we aim for a fully global module, we always set $L$ to $\operatorname{max}(h, w)$ and make the position branch global.
>
> The significance of global attention networks over local attention alternatives is three-fold:
>
> 1. Our global attention approach brings better performance-cost trade-off than all existing local attention networks, as Table 2 shows. GSA-Net achieved the best accuracies using the least resources. Additionally, note the GSA-Net uses a fairly standard ResNet baseline. Many of the the competing approaches have the exact same 76.9% top-1 ResNet-50 baseline (e.g. SASA (Ramanchandran et al., 2020), SAN (Zhao et al., 2020)).
> 2. Global attention is much faster than local attention. The reason is that global attention utilizes standard NN operations (e.g. matrix multiplication), while local attention needs specialized primitives. Those primitives are very hard to optimize on modern accelerators (e.g. GPUs, TPUs). Table 2 clearly shows that we are ~4 times faster than SASA (Ramanchandran et al., 2020). SAN (Zhao et al., 2020) did not report their inference speed or release benchmarkable code, but we believe SAN and any other local attention networks would not be significantly faster than SASA.
> 3. An additional benefit of using standard operators is that global attention is significantly easier to implement. Our code shows this. (It is currently pending internal approval for submission. We hope it will be approved before the end of the discussion phase.)

---

> ### Author Response · Authors · 2020-11-25
> **Additional response regarding importance of globality**
>
> We present additional experimental results on the importance of globality.
>
> | Spatial extent | Model | Top-1 acc. | Top-5 acc. | Parameters (M) | FLOPs (G) | Latency (ms) |
> | -: | - | -: | -: | -:| -: | -: |
> | 3 | ResNet-50 | 76.9 | 93.5 | 25.6 | 8.2 | 22.6 |
> | | SASA-ResNet-50 (Ramanchandran et al., 2020) | 76.4 | - | 18.0 | 6.6 | 143.1  |
> | | **GSA-ResNet-50** | 77.8 | 93.6 | 18.0 | 7.2 | 31.2 |
> | 7 | SASA-ResNet-50 (Ramanchandran et al., 2020) | 77.4 | - | 18.0 | 7.0 | 131.9 |
> | | **GSA-ResNet-50** | 78.4 | 93.9 | 18.0 | 7.2 | 31.2 |
> | $\infty$ | **GSA-ResNet-50** | 78.5 | 93.9 | 18.1 | 7.2 | 31.7 |
>
> As the table shows, increasing the spatial extent leads to steady gain of accuracy. Although a spatial extent of 7 already gives most of the accuracy benefits of a global spatial extent, going from 7 to global also incurs negligible costs for GSA-Net. This is because we implement *local* variants of GSA-Net by masking positional logits beyond the spatial extent (e.g. a 7x7 window), which does not save any computation. (It marginally saves parameters since it requires fewer positional embeddings.) If we alternatively choose to actually implement local attention similar to SASA (Ramanchandran et al., 2020), it saves some theoretical computation but is practically substantially slower due to the use of primitives that are highly accelerator-unfriendly.
>
> Note that comparing the accuracy of GSA-Net and SASA at different spatial extents might be unfair for SASA since for local GSA-Nets, only the positional branch is local, while the content branch is still global. The reason is that local attention does not allow our $\boldsymbol{Q}\left(\boldsymbol{K}^\mathsf{T}\boldsymbol{V}\right)$ computational order. If we want to implement a local content branch, we would have to adopt SASA's implementation, which would make us as slow as SASA. Therefore, the comparison against SASA is purely to show that masking is the fastest implementation for local attention. Therefore, going global with GSA-Net is  essentially free.

---

### Decision · Program_Chairs · 2021-01-07
**Final Decision**

**Decision:**

Reject

**Comment:**

Paper was reviewed by four expert reviewers. Unfortunately all reviewers, uniformly felt that paper fell marginally bellow bar and argue for rejection. A number of concerns have been identified by the reviewers in the review phase.  Those included: (1) lack of novelty [Reviewer3, Reviewer4], (2) lack of various ablations [Reviewer 2], (3)  issues with experimental setup [Reviewer4], and (4)  lack of significant improvements in performance [Reviewer 1, Reviewer 3]. While authors addressed some of the concerns with provided experiments and ablations during the rebuttal, Reviewers remained unconvinced on the main concerns of novelty and significance. As such, the reviewers are unanimous in their assessment and AC does not see a reason to overturn this consensus.